# A Randomized Controlled Trial on *Pleurotus eryngii* Mushrooms with Antioxidant Compounds and Vitamin D_2_ in Managing Metabolic Disorders

**DOI:** 10.3390/antiox11112113

**Published:** 2022-10-26

**Authors:** Stamatia-Angeliki Kleftaki, Charalampia Amerikanou, Aristea Gioxari, Dimitra Z. Lantzouraki, George Sotiroudis, Konstantinos Tsiantas, Thalia Tsiaka, Dimitra Tagkouli, Chara Tzavara, Lefteris Lachouvaris, Georgios I. Zervakis, Nick Kalogeropoulos, Panagiotis Zoumpoulakis, Andriana C. Kaliora

**Affiliations:** 1Department of Nutrition and Dietetics, School of Health Science and Education, Harokopio University, 70 El. Venizelou Ave., 17676 Athens, Greece; 2Department of Nutritional Science and Dietetics, School of Health Science, University of the Peloponnese, Antikalamos, 24100 Kalamata-Messinia, Greece; 3Institute of Chemical Biology, National Hellenic Research Foundation, 48, Vas. Constantinou Ave., 11635 Athens, Greece; 4Department of Food Science and Technology, University of West Attica, Ag. Spyridonos, 12243 Egaleo, Greece; 5Dirfis Mushrooms P.C., Kathenoi, 34018 Euboea, Greece; 6Laboratory of General and Agricultural Microbiology, Department of Crop Science, Agricultural University of Athens, 11855 Athens, Greece

**Keywords:** *Pleurotus eryngii*, metabolically unhealthy, antioxidants, glucose levels, inflammation, oxidative stress, vitamin D, physical health

## Abstract

This study examined the effects of a *Pleurotus eryngii* mushroom snack on metabolically unhealthy patients. After harvest, mushrooms were baked and subjected to UV-B irradiation to enhance vitamin D_2_ content. A randomized controlled trial was conducted for three months with two arms. Both groups received conventional nutritional counseling for metabolic disorders, while the intervention group had to consume the snack daily as well. We collected blood samples at the beginning and the end of the study to determine biochemical measurements and serum 25(OH)D2 and to evaluate inflammation and oxidative stress. One hundred patients consented and were randomized. Comparatively to the control group, snack consumption regulated glucose levels and reduced body weight, fat, waist and hip circumferences. In addition, 25(OH)D2 increased significantly in the intervention group. The levels of LDL and SGOT were lower only in the intervention group. Levels of IL-6 and ox-LDL decreased in the mushroom group, while the overall physical health increased. These findings suggest potential antidiabetic, antiobesity, anti-inflammatory and antioxidant health benefits of the snack to metabolically unhealthy individuals.

## 1. Introduction

Obesity is a chronic relapsing disease, characterized by excessive body fat accumulation. It is associated with a series of disorders, which are threatening for the global public health and economy [1]. During the last few decades, there has been a significant global increase in obesity rate. According to WHO, the prevalence of overweight and obesity cases in the adult population is 39% and 13%, respectively [2]. In recent years, there has been a growing interest in metabolic disorders that represent a cluster of abnormalities, including abdominal adiposity, dyslipidemia, glucose intolerance, insulin resistance and hypertension [3]. There are many health and economic consequences associated with obesity and metabolic disorders, such as type 2 diabetes mellitus (T2DM), cardiovascular disease (CVD), nonalcoholic fatty liver disease (NAFLD), certain types of cancer, obstructive sleep apnea and depression [4,5].

In order to address the obesity epidemic, a series of approaches have been proposed, including changes in diet. More specifically, there is an increasing interest in the consumption of functional foods which may improve cardiometabolic health [6]. In this respect, edible mushrooms may be a great choice since they demonstrate various medicinal and functional properties [7,8]. Pleurotus species are among the most widely cultivated mushrooms and account for approximately 20% of the respective total global production [9]; they are rich in proteins, fibers, essential amino acids, carbohydrates, water-soluble vitamins, and minerals [10]. In particular, the fruitbodies of *Pleurotus eryngii (P. eryngii)*, also known as “King Oyster mushroom”, present a high content of bioactive compounds, such as ergosterol, beta-glucan, and ergothioneine [11,12]. In addition, they are a good source of vitamin D_2_, which is produced by the conversion of ergosterol to ergocalciferol after exposure to UV light [13]. As a result, *P. eryngii* exhibits a variety of pharmacological activities and important nutritional value, and has gained a great deal of research attention in recent years due to its antioxidant, immunoregulatory, antihyperlipidemic, and other activities [14].

Although there is a variety of literature on the cardioprotective activities of *P. eryngii* extracts in vitro, as well as in animal studies [15,16,17,18,19], no human clinical trials exist in this field. Our interest in the potential benefits of *P. eryngii* on metabolic parameters was further reinforced by our recent study that showed that a meal with *P. eryngii* can ameliorate postprandial glycemia and appetite, and can regulate ghrelin levels at the postprandial state [20].

Thus, the aim of the present study was to investigate the effects of a *P. eryngii* mushroom snack administered daily for 3 months in metabolically unhealthy patients as regards the improvement in biochemical, inflammatory, oxidative stress, and lifestyle parameters.

## 2. Materials and Methods

### 2.1. Study Design

The protocol was reviewed and approved by the Ethics Committee of Harokopio University (ID protocol: 62/03-07-2018). The trial was conducted in accordance with the Helsinki Declaration and the Data Protection Act 1998 and was registered with clinicaltrials.gov (ID Number: NCT04081818). Eligible subjects were enrolled in the study after being informed in detail about its nature and all procedures and having given their written consent for participation. The study took place in Harokopio University of Athens, Greece in 2020 and 2021.

One hundred and eighty metabolically unhealthy subjects were recruited according to predetermined inclusion and exclusion criteria in a randomized controlled trial (Figure 1). Eligible subjects were between the ages of 18 and 76 years and diagnosed with metabolic disorders. They also had a stable weight for at least 3 months before enrollment and a moderately active lifestyle. Exclusion criteria included pregnancy and lactation, untreated thyroid disease, any use of supplements within 3 months pre-intervention, a history of drug and/or alcohol abuse, and psychiatric or mental disorders.

After giving their consent, participants were randomized to the control or intervention group for 3 months. Both arms were given standard nutritional counseling for metabolic disorders throughout the 3-month trial, whereas the intervention group had also to consume the snack daily for 3 months. Randomization was carried out by an independent biostatistician, and compliance was monitored biweekly through phone calls. All baseline assessments were repeated at follow-up.

### 2.2. Snack Preparation

Following harvest, mushrooms were initially cut into 2 mm-thick slices; then, yeast extract (as a flavor enhancer) and garlic flavor powder were added at a ratio of 0.5% (***w/w***), and the slices were baked at 120 °C for 20 min in a professional oven. After baking, the sliced mushrooms were subjected to UV-B irradiation (290–315 nm; 39 W lamps positioned at 20 cm from one another; mushrooms were placed at a distance of 20 cm from the light source, and were subjected to illumination for 120 min) to enhance vitamin D2 content. For the intervention, sachets made of aluminum foil were filled with the generated mushroom product (‘snack’, 6 g in each sachet) and were hermetically sealed. The nutritional composition and caloric content of the snack are presented in Table 1, while the glucan content was ≈2.5 g [12]. To ensure food safety, microbiological tests that are necessary for the food sector, including both control of safety and of hygiene criteria, as the applicable law requires them (Regulation 2073/2005), were carried out in an accredited laboratory. Finally, an experienced panel (Laboratory of Food Chemistry and Technology, School of Chemical Engineering, National Technical University of Athens) carried out the sensory testing of the mushroom snack (data not shown).

### 2.3. Outcomes

The primary outcome of the study was the changes in insulin sensitivity, and more specifically changes in fasting glucose levels after the 3 months’ intervention.

Secondary outcomes included changes in vitamin 25(OH)D2 levels, anthropometric measures, biochemical parameters, inflammatory and oxidative stress markers, as well as changes in quality of life.

### 2.4. Medical, Dietary, and Quality-of-Life Assessment

Detailed medical history was obtained including personal, family, medical history, and medication.

Dietary intake was assessed using a 24 h recall record (four non-consecutive days of the week) and was analyzed using Nutritionist Pro™ (Axxya Systems, Stafford, TX, USA) software (version 7.1.0).

Physical activity level was evaluated via the International Physical Activity Questionnaire Short Form (IPAQ-SF). This 7-day recall instrument measures frequency and duration of walking, moderate, and vigorous physical activity [21].

Self-esteem was evaluated via the Rosenberg Self-Esteem scale. It includes a 10-item scale that estimates global self-worth by measuring both positive and negative feelings about the self. There is a 4-point rating scale (1 to 4) from strongly agree to strongly disagree. The total score ranges from 10 to 40 points. Higher scores indicate higher self-esteem [22].

As obesity is considered a risk factor for depressive disorders, the 10-item questionnaire Center for Epidemiologic Studies Depression Scale Revised (CESD-R-10) was applied pre- and post-intervention. Subjects scoring ≥16 (range 0–60) are considered at risk for prevalent depression [23].

Additionally, the insomnia level was evaluated via the Athens Insomnia Scale (AII) which records the assessment of any sleep difficulty. It consists of eight items that assess nocturnal sleep problems and daytime dysfunction. A higher score indicates greater severity of insomnia symptoms [24].

Finally, the subjects completed questionnaires regarding their physical and mental health (Short Form-12 Physical Composite Score (PCS-12) and Mental Composite Score (MCS-12). The questionnaires include 12 questions with dichotomous responses (yes/no), ordinal (excellent to poor), or expressed by a frequency (always to never). The higher the score, the better the health status [25].

### 2.5. Anthropometric Measurements

Body weight, body fat, fat free mass (FFM), total body water (TBW), and visceral fat rating were measured with bioelectrical impedance analysis (Tanita BC-418, Tokyo, Japan). Height was measured using a stadiometer (Seca Mode 220, Hamburg, Germany) with subjects not wearing shoes, their shoulders in a relaxed position, and their arms hanging freely. Waist circumference (WC) was determined at the midpoint between the lower margin of the last palpable rib and the top of the iliac crest in a standing position at the end of gentle expiration. Hip circumference (HP) measurement was taken around the widest portion of the buttocks. Body mass index (BMI) was computed as weight (kg)/height (m)^2^.

### 2.6. Blood Collection

Blood samples (20 mL) were collected after an overnight fast for biochemical and laboratory analyses. Blood samples were centrifuged at 3000 rpm for 10 min at 20 °C for plasma and serum isolation. EDTA was used as an anticoagulant for plasma isolation. All samples were stored at −80 °C until further laboratory analysis.

### 2.7. Laboratory Analyses

#### 2.7.1. Biochemical Analyses

Serum glucose, insulin, urea, uric acid, creatinine, total cholesterol (TC), HDL-C, low-density lipoprotein cholesterol (LDL-C), triglycerides (TG), alanine aminotransferase (ALT), aspartate aminotransferase (AST), γ-glutamyl transferase (γ-GT), alkaline phosphatase (ALP), uric acid, lactate dehydrogenase (LDH), iron (Fe), ferritin, albumin, C-reactive protein (CRP) were measured in serum with an automatic biochemical analyser (Cobas 8000 analyser, Roche Diagnostics GmbH, Mannheim, Germany).

#### 2.7.2. Evaluation of Inflammation and Oxidative Stress

Interleukin-6 (IL-6), tumor necrosis factor α (TNFα), leptin, adiponectin (R&D Systems, Inc., Minneapolis, MN, USA), MPO (Thermo Fisher Scientific Inc., Waltham, MA, USA), oxLDL (Mercodia, AB, Uppsala, Sweden) and 8-isoprostanes (Abcam, Cambridge, UK) were measured applying ELISA as indicators of chronic inflammatory grade and oxidative stress. All ELISA measurements were conducted in duplicate.

#### 2.7.3. 25(OH)D2 and 25(OH)D3 Using Liquid Chromatography–Tandem Mass Spectrometry (LC-MS/MS)

##### Reagents and Standards

Standards of 25-Hydroxy vitamin D_2_, 25-Hydroxy vitamin D_3_ as well as vitamin D_2_ deuterated (25-Hydroxy Vitamin D_2_-d6), used as internal standard (IS) were purchased from Santa Cruz Biotechnology (Dallas, TX, USA), while D_3_ deuterated (25-Hydroxy Vitamin D_3_-d6, IS) was acquired from Glentham Life Sciences (Leafield, UK). All standard stock solutions were prepared in methanol and stored at −18 °C. All solvents were of a liquid chromatography–mass spectrometry (LC-MS) grade. More specifically, acetonitrile and formic acid were provided from Carlo Erba (Reuil, France), whereas water, hexane and methanol were purchased from Fischer Scientific (Hampton, VA, USA) and Sharlau (Barcelona, Spain), respectively.

##### 25(OH)D2 and 25(OH)D3 Extraction Procedure

The extraction of vitamins 25(OH)D2 and 25(OH)D3 was performed as described in a previously published work with slight modifications [26,27,28]. Briefly, frozen serum samples (280 μL) were thawed, mixed gently (multi vortex V-32, BioSan, Riga, Latvia) with another 280 μL acetonitrile containing 0.1% formic acid and finally spiked with 40 μL of a combined mixture of the internal standards’ solution at concentrations 0.5 and 4.0 μg mL^−1^ of 25(OH)D2-d6 and 25(OH)D3-d6, respectively. The mixture was incubated for 30 min in 7 °C in order to induce protein precipitation. Then, 1200 μL of hexane was added to the above solution followed by a 5 min vortex. The new solution was incubated for another 25 min at 7 °C and then centrifuged at 10 °C and 12,000 rpm (Centrifuge Z32 HK, Hermle, Wehingen, Germany). After centrifugation, 900 μL of the upper organic phase was transferred to a new Eppendorf tube. Next, 1000 μL of hexane was added to the remaining solution and treated as previously described (vortex, incubation, centrifugation and collection of 1000 μL of the supernatant). The merged organic layers (1900 μL total) were centrifuged for 15 min at 10 °C and 1700 μL of the supernatant was collected in order to remove the solvent by using a nitrogen pump. Prior to analysis, the dry residue was reconstituted using a mixture of 2-propanol (35 μL) and methanol (50 μL), followed by a 10 min centrifugation (12,000 rpm at 10 °C). Finally, the supernatant was inserted in Liquid chromatography–mass spectrometry (LC-MS) vials.

##### 25(OH)D2 and 25(OH)D3 Analysis

LC-MS analysis was used for the identification and quantification of 25(OH)D2 and 25(OH)D3. LC-MS included an API 3200 QTrap triple quadrupole/Linear ion trap mass spectrometer (AB Sciex, Foster City, CA, USA) coupled to an Agilent 1200 HPLC system (Agilent, Waldbronn, Germany). All spectra were processed by the Analyst software (version 1.4.2, AB Sciex, Foster City, CA, USA). In addition, for the separation of the four analytes, a Poroshell HPH-C18 column was used (2.1 particle size, 50 mm i.d., 2.7 μm). The elution of the analytes was performed by using a gradient system with two solvents (Solvent A, water with 0.1% formic acid and Solvent B: methanol with 0.1% formic acid) at flow rate 0.150 mL min^−1^. More specifically, the gradient started with 23% of solvent A, reduced to 0% over 8 min of analysis, and, in 23 min, percentage of solvent A ramped to initial conditions (23%), which remained till the end of the analysis. The injection volume was set at 5 μL. The MS/MS parameters for all the analytes were optimized with the direct infusion of a mixed standard methanol solution (1 mg mL^−1^) of the analytes. Tandem mass spectrometry analysis of all samples was performed in a positive mode using an electrospray chemical ionization (ESI) source.

The identification and quantification of the two forms of vitamin D were based on the fragmentation of the precursor ions into the respective product ions using a multiple monitoring reaction (MRM) technique. More specifically, product fragments of deuterated 25-Hydroxy vitamin D2 (*m/z* 419.3 > 355.4) were observed at a retention time (RT) of 12. 86 min, while product fragments of 25-Hydroxy vitamin D2 (*m/z* 413.4 > 355.4) were identified at a retention time of 12.91 min. Similarly, a product ion of 25-Hydroxy D3 (*m/z* 401.4 > 365.3) was detected at a retention time of 12.63 min, while a product ion of deuterated 25-Hydroxy D3 (*m/z* 407.5 > 371.4) was identified at a retention time of 12.68 min.

A mixture of plasma samples (pooled), which contained traces of the investigated analytes and 20 uL of each internal standard (25(OH)D2-d6 and 25(OH)D3 -d6), was used for the construction of calibration curves. The developed method exhibited good linearity (Peak area = 5.2456(±0.1362) × C 25(OH)D2 + 0.0004(±0.0130), R2 = 0.995 and Peak area = 8.2914(± 0.3029) × C 25(OH)D3–0.028(± 0.031), R2 = 0.993, respectively) within a wide range of concentrations (0.0001 to 0.25 μg mL^−1^, n = 10). The method showed also good (a) intra-day (repeatability) (4.28% and 4.0%, for 25(OH)D2 and 25(OH)D3, respectively) and inter-day precision (reproducibility) (10.28% and 9.51%, for 25(OH)D2 and 25(OH)D3, respectively), (b) accuracy (115.1% and 86.42%, for 25(OH)D2 and 25(OH)D3, respectively) and (c) recovery (95.38% and 73.43%, for 25(OH)D2 and 25(OH)D3, respectively), as calculated by the *Official Journal of the European Communities* guidelines [29]. The limit of detection (LOD) was determined at 0.29 ng mL^−^^1^ for 25(OH)D2 and at 0.48 ng mL^−^^1^ for 25(OH)D3, whilst the limit of quantification (LOQ) was 0.00095 μg mL^−^^1^ and 0.00158 μg mL^−^^1^ for 25(OH)D2 and 25(OH)D3, respectively.

### 2.8. Sample Size Determination and Statistical Analysis

Continuous variables are presented with mean and standard deviation (SD). Quantitative variables are presented with absolute and relative frequencies. All analyses were conducted on an intention-to-treat basis. For the comparison of proportions, chi-square and Fisher’s exact tests were used. For the comparison of means between the control and intervention group, Student’s t-test was computed. To reduce the bias implicit in utilizing only complete cases, multiple imputation procedures for all data were implemented. Differences in changes of study variables during the follow-up period between the two study groups were evaluated using repeated measurements analysis of variance (ANOVA). All *p* values reported are two-tailed. Statistical significance was set at 0.05 and analyses were conducted using SPSS statistical software (version 24.0, IBM, New York, NY, USA).

A repeated-measures power analysis was conducted for a single within-subjects factor assessed over two time points. For this design, 50 participants per group achieve a power of 0.94 for the between-subjects main effect at an effect size of 0.30; a power of 0.95 for the within-subjects main effect at an effect size of 0.20; and a power of 0.95 for the interaction effect at an effect size of 0.20.

## 3. Results

### 3.1. Subject Characteristics

A total of 100 participants (35 males, 65 females) were randomized in two equally sized groups. Participant characteristics are shown in Table 2 and were similar in both groups.

No side effects were reported. Additionally, as recorded through biweekly phone calls from experienced dieticians, the overall protocol compliance was >80%.

### 3.2. Effect on Biochemical Indices

Regarding the biochemical measurements (Table 3), glucose decreased only in the intervention group after 3 months, while it remained unchanged in the control group, with the degree of change being significantly different between the two groups. LDL and SGOT decreased only in the intervention group and the between group differences changes were not significant. Additionally, glucose levels were different between groups in post-intervention as well as albumin levels in pre- as well as in post-intervention, but no other significant differences were observed in albumin.

### 3.3. 25(OH)D2 and 25(OH)D3 Levels

As for 25(OH)D2 and 25(OH)D3 (Table 4), no significant differences occurred at baseline and at follow-up between the two groups. During follow-up, 25(OH)D2 increased significantly only in the intervention group. Consequently, the degree of change of 25(OH)D2 differed significantly between the two groups. No significant time differences were found in 25(OH)D3.

### 3.4. Effect on Anthropometric Characteristics

Table 5 presents changes in anthropometric characteristics after 3 months in both groups. Weight, fat (kg and %), BMI, waist and hip circumferences decreased only in the intervention group after 3 months, while they remained unchanged in the control, with the mean changes being significantly different between the two groups. When comparing pre- and post- levels in the groups, no significant changes were observed. Only TBW was lower in the intervention group than in the control group at follow-up, without the differences in mean changes being significant.

### 3.5. Effect on Inflammatory and Oxidative Stress Biomarkers

Changes in inflammatory and oxidative stress biomarkers are presented in Table 6. IL-6 and oxLDL decreased in the intervention group after 3 months and remained unchanged in the control group, with the mean changes being significantly different between the two groups. At baseline and at follow-up, both groups had similar values.

### 3.6. Effect on the Quality of Life

Finally, regarding quality of life, changes after the intervention are presented in Table 7. Physical health score was significantly higher in the intervention group at follow-up and the degree of change after 3 months differed significantly between the two groups. Additionally, PCS-12 and physical activity (total MET- min/week) were higher in the intervention group and pre- and post-intervention accordingly. No significant changes were observed in the quality-of-life scores that derive from all the other questionnaires.

## 4. Discussion

During the last few decades, functional ingredients of foods have attracted research interest in the prevention and management of obesity and related metabolic disorders. Several bioactive compounds of mushrooms have been documented to have beneficial impacts on various metabolic markers.The consumption of mushrooms has been associated with cardioprotective effects, such as hypocholesterolemic, antihyperglycemic, antihypertensive, anti-inflammatory and antioxidant properties [30]. To the best of our knowledge, this is the first randomized controlled clinical trial exploring the effect of vitamin D2-enhanced *P. eryngii* snack on parameters related to metabolic disorders.

We succeeded in proving our primary hypothesis that the daily consumption of the snack regulates glucose levels compared with the control group. Our findings are in accordance with the existing literature, as it has been shown that mushrooms possess an antidiabetic effect mainly due to their polysaccharide content [31] and by increasing glucokinase activity [32]. *P. ostreatus* exhibits similar effects, with a 7-day consumption of a cooked mushroom meal in exchange for vegetables to hospitalized patients with insulin resistance, resulting in a 22% reduction in fasting glucose [33].

During follow-up, 25(OH)D2 levels increased significantly only in the intervention group with the degree of change being significantly different between the intervention and the control group. No significant differences were found in 25(OH)D3. Recently, Hu et al. showed that UV irradiation increases vitamin D2 concentration in *P. ostreatus* mushrooms in ethanol suspension, thus enhancing their nutritional value [34]. Several studies have demonstrated that UV irradiation of edible mushrooms results in a high rate of ergosterol conversion to vitamin D2; in addition, serum 25(OH)D levels are increased as shown in some clinical trials [35]. Other clinical trials showed that serum 25(OH)D did not significantly increase by consumption of UV-exposed mushrooms [36]. Although most studies have investigated D3 supplementation and its benefits on human health, D2 has also been shown to exhibit beneficial effects. Hence, D2 improves the quality of life in osteoarthritis subjects [37], and it regulates endothelial function [38] and arterial stiffness [39]. Endothelial dysfunction is not only a consequence of insulin resistance but also impairs insulin signaling to further reduce insulin sensitivity, thereby resulting in a destructive cycle in metabolic disorders and diabetes [40].

Additionally, participants of the intervention group exhibited a reduction in body weight, fat, waist, and hip circumference, whereas no significant alterations appeared in the control group, with the mean changes being different between the two intervention groups. This may be due to better appetite regulation as also shown in our previous study where *P. eryngii* mushrooms ameliorate appetite and suppress ghrelin levels postprandially owed to their beta-glucan content [20]. Moreover, it has been shown that a meal enriched with powder from dried oyster mushrooms can increase GLP-1 postprandially and decrease hunger rate for the same reasons [41]. In a long-term (1 year) clinical trial with obese patients that substituted mushrooms for red meat lower BMI and waist circumference were reported [42].

The results of the present investigation suggest that *P. eryngii* might improve lipid profile. LDL levels decreased only in the intervention group although mean changes were not different between the intervention and the control group. Polysaccharides in mushrooms, including chitosans and glucans can reduce LDL levels [43,44]. However, modest weight loss in obese individuals may also provide lower fasting glucose and LDL levels [45]. Our findings are in consistency with those of Choudhury et al. [46] who showed a reduction in TC and LDL-C levels of obese hypertensive non-diabetic males administered with 3 g of *P. ostreatus* powder in capsule form daily for 3 months. Similar results were also found in another study where lipid levels, including TC and LDL-C were significantly different after the consumption of *Agaricus bisporus* cooked with olive oil [47]. In contrast, other studies reported that LDL-C remains unchanged [48,49].

As for SGOT/AST levels, consumption of several mushroom species, including those of the genus Pleurotus, can reduce AST levels according to results obtained from animal models [50]. Pleurotus species can generate the paths for diffraction of different liver enzymes and reduce the levels of serum enzyme activities [51].

As obesity is associated with chronic low-grade inflammation and oxidative stress, studying them in clinical trials of obese and metabolic patients is essential. Insulin signaling is impaired and chronic inflammation is induced by such markers [52]. In the present, a decrease in IL-6 and ox-LDL was found in the mushroom group, which did not appear in the control, the mean changes being significantly different between groups. To the best of our knowledge, this study is the first to demonstrate that *P. eryngii* mushrooms regulate inflammation and oxidative stress in humans. The CRP alterations after mushroom intake were not accompanied by IL-6 and oxLDL differences when compared with the control [53]. Similarly, other trials failed to report any significant effects on inflammatory markers [54,55]. The presence of lovastatin, a member of the statins family that lower TC and LDL levels and reduce the risk of coronary heart disease [56] and the antioxidants ergothioneine [11] and selenium [57] have been detected in relatively high amounts in *P. eryngii*. Added to the above, the increase in vitamin D2 content via UV radiation has been shown to ameliorate inflammation in humans [58,59]. However, when examining the effects of such complex matrices on metabolic health, it should be emphasized that the overall activity may be due to the synergistic effects of all compounds rather than the individual activities of specific constituents. DPPH and FRAP experiments in the extracts from the snack have shown a noticeable antioxidant capacity (399.92 ±10.14 μmol Trolox equivalents/100 g and 16.31 ± 0.07 μmol ascorbic acid equivalents/100 g, respectively). This may be due to the remaining antioxidant content in mushrooms, but also due to the antioxidants that arise from baking (i.e., the Maillard reaction products) and contribute to the overall antioxidant capacity.

With regard to the quality of life, PCS-12 was significantly higher in the intervention group at follow-up with a significant difference in the degree of mean change between the two groups. PCS-12 is a very good marker for explaining variations in the quality of life across BMI and seems to be lower in obese patients compared to normal-weight controls [60]. In our study, the 4.1-point increase in PCS-12 in the intervention group may be explained by the regulation of several metabolic parameters, such as glucose, weight, BMI, fat, and of course by the overall improvement in the inflammatory and oxidative stress status. Additionally, the quality of life may have been improved due to an improvement in the quality of sleep and, more specifically, in snoring (snoring was stopped successfully in 23% of the participants at follow-up in the intervention group, whereas no participant stopped snoring in the control group).

Overall, the results of our randomized controlled clinical trial should be viewed in light of the fact that the trial could not be blinded; thus, a degree of bias is inevitable. However, we believe that the above is counterbalanced by several strengths, such as the adequate power of the study, and the satisfying degree of compliance, as verified by consistent phone calls with the participants, as well as with follow-up increased levels of vitamin 25(OH)D2 only in the intervention group. Finally, another strength was the very careful selection of the participants according to tight inclusion and exclusion criteria, as well as the successful randomization process mitigating possible bias in the study.

## 5. Conclusions

In conclusion, the consumption for 3 months of a snack prepared from *P. eryngii* mushrooms with enhanced content of vitamin D_2_ and with other bioactive compounds was associated with a significant reduction in glucose, body weight, BMI, and body fat, in parallel with an increase in serum 25(OH)D2 and quality of life. Additionally, the snack resulted in an improved profile of inflammatory and oxidative stress status. Overall, these findings suggest potential antidiabetic, antiobesity, anti-inflammatory, and antioxidant health benefits of the snack to metabolically unhealthy individuals. In light of the increase in the obesity epidemic and the resulting metabolic disorders, such data are considered of great importance. However, essential information about the synergistic activity of the components of *P. eryngii* that are beneficial to metabolic health is needed to exploit further the value of the results.

## Figures and Tables

**Figure 1 antioxidants-11-02113-f001:**
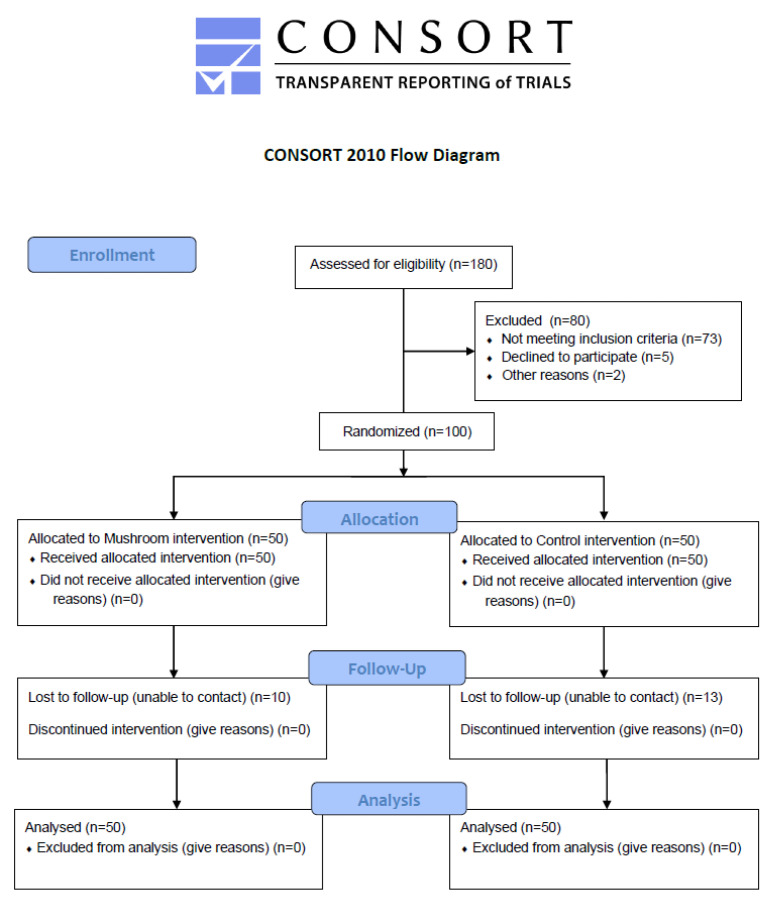
Study Flow diagram.

**Table 1 antioxidants-11-02113-t001:** Nutritional composition of the snack.

	Snack (6 g)
Energy content (Kcal)	18.42
Available carbohydrates (g)	1.77
Fat (g)	0.288
Protein (g)	1.35
Salt (g)	0.08
Vitamin D_2_ (μg)	20

**Table 2 antioxidants-11-02113-t002:** Demographics in control and intervention groups. The results are given as N (%) of the total number.

		Group		*p*
		Control	Intervention	
		Ν (%)	Ν (%)	
Gender	Men	21 (42)	14 (28)	0.142 +
	Women	29 (58)	36 (72)	
Age. mean (SD)		53.7 (10.8)	54.9 (11.8)	0.610 ‡
Nationality	Greek	48 (96.0)	48 (98.0)	>0.999 +
	Other	2 (4.0)	1 (2.0)	
Family status	Married	39 (78)	38 (77.6)	>0.999 ++
	Divorced	2 (4)	3 (6.1)	
	Single	5 (10)	5 (10.2)	
	In a relationship	2 (4)	1 (2)	
	Widowed	2 (4)	2 (4.1)	
Children	0	16 (32)	9 (18.4)	0.376 +
	1	8 (16)	7 (14.3)	
	2	19 (38)	26 (53.1)	
	3	7 (14)	7 (14.3)	
Educational level	None	1 (2)	0 (0)	0.482 ++
	Primary	0 (0)	2 (4.1)	
	Secondary	19 (38)	20 (40.8)	
	University	20 (40)	21 (42.9)	
	Other	10 (20)	6 (12.2)	
Years of education, mean (SD)		15.3 (3.1)	14.7 (3.8)	0.436 ‡
Occupation	Full time in public sector	26 (52)	20 (40.8)	0.071 ++
	Full time in private sector	11 (22)	4 (8.2)	
	Part time in public sector	0 (0)	2 (4.1)	
	Part time in private sector	0 (0)	1 (2)	
	Freelancer	4 (8)	4 (8.2)	
	Household	1 (2)	6 (12.2)	
	Unemployed	0 (0)	1 (2)	
	Pensioner	8 (16)	11 (22.4)	
Menstruation ^1^	No	18 (62.1)	24 (70.6)	0.475 +
	Yes	11 (37.9)	10 (29.4)	
Overweight	No	0 (0)	1 (2)	0.495 ++
	Yes	50 (100)	48 (98)	
Hypertension	No	13 (26)	8 (16)	0.220 +
	Yes	37 (74)	42 (84)	
Diabetes	No	34 (68)	28 (56)	0.216 +
	Yes	16 (32)	22 (44)	
Dyslipidemia	No	5 (10)	6 (12)	0.749 +
	Yes	45 (90)	44 (88)	
Smoking	Daily	10 (20)	9 (18)	0.949 ++
	Occasionally	1 (2)	2 (4)	
	Daily for stopped smoking	15 (30)	12 (24)	
	Occasionally but stopped smoking	3 (6)	3 (6)	
	No	21 (42)	24 (48)	
Alcohol consumption	Daily	1 (2)	2 (4.1)	0.544 ++
	4–6 times/week	2 (4)	3 (6.1)	
	1–3 times/week	14 (28)	13 (26.5)	
	1–2 times/month	13 (26)	6 (12.2)	
	Rarely	13 (26)	14 (28.6)	
	Never	7 (14)	11 (22.4)	

^1^ only in women; + Pearson’s chi-square test; ++ Fisher’s exact test; ‡ Student’s *t*-test.

**Table 3 antioxidants-11-02113-t003:** (a): Changes in biochemical measurements after 3 months in control vs. the intervention group; (b): Changes in lipid profile, hepatic enzymes and markers of renal function after 3 months in control vs. the intervention group.

(a)						
		Pre	Post	Change		
	Group	Mean (SD)	Mean (SD)	Mean (SD)	P2	P3
Glucose (mg/dL)	Control	104 (34.7)	107 (37)	3 (19.3)	0.283	**0.001**
	Intervention	101.6 (26.8)	90.7 (15.4)	−10.9 (20.2)	**<0.001**	
	P1	0.704	**0.005**			
Insulin (μIU/mL)	Control	16.7 (12.1)	15.9 (11.4)	−0.9 (12.9)	0.617	0.915
	Intervention	18.8 (13.8)	17.6 (10.2)	−1.1 (11.7)	0.519	
	P1	0.444	0.425			
ALP (U/L)	Control	67.1 (17.8)	70 (22.7)	2.9 (13.6)	0.110	0.059
	Intervention	73.3 (21.5)	71.4 (24.6)	−2 (11.4)	0.279	
	P1	0.122	0.775			
Fe (μg/dL)	Control	82.5 (31)	81 (30.6)	−1.5 (35)	0.772	0.261
	Intervention	78.2 (28.4)	84.9 (30)	6.7 (37)	0.196	
	P1	0.471	0.529			
Ferritin (ng/mL)	Control	91.6 (106.2)	101.7 (104.4)	10.1 (41.3)	0.165	0.119
	Intervention	98.8 (81.6)	92.7 (83.7)	−6 (59.2)	0.410	
	P1	0.709	0.638			
Albumin (g/dL)	Control	4.35 (0.29)	4.39 (0.25)	0.04 (0.32)	0.359	0.180
	Intervention	4.62 (0.28)	4.58 (0.35)	−0.04 (0.31)	0.327	
	P1	**<0.001**	**0.003**			
CRP (mg/L)	Control	4.94 (5.81)	5.46 (4.89)	0.52 (5.23)	0.411	0.906
	Intervention	4.29 (4.04)	4.91 (4.45)	0.63 (3.48)	0.327	
	P1	0.518	0.561			
LDH (U/L)	Control	179.6 (125.8)	161.9 (38.8)	−17.7 (118.9)	0.176	0.603
	Intervention	161.6 (40.7)	153.5 (42.2)	−8.1 (51.5)	0.540	
	P1	0.341	0.304			
**(b)**						
		**Pre**	**Post**	**Change**		
	**Group**	**Mean (SD)**	**Mean (SD)**	**Mean (SD)**	**P2**	**P3**
ΤC (mg/dL)	Control	186.3 (32.8)	182 (40)	−4.3 (40.7)	0.419	0.643
	Intervention	200.3 (50.9)	196.6 (55.2)	−3.7 (33.5)	0.485	
	P1	0.105	0.134			
TG (mg/dL)	Control	143.5 (93.1)	134.6 (81.2)	−9 (73.4)	0.440	0.921
	Intervention	162 (94.1)	154.6 (82.3)	−7.3 (89.7)	0.532	
	P1	0.330	0.225			
HDL (mg/dL)	Control	47.1 (11.6)	48.0 (12.8)	0.9 (6.9)	0.327	0.072
	Intervention	51 (9.6)	49.6 (9.8)	−1.4 (5.2)	0.117	
	P1	0.074	0.479			
LDL (mg/dL)	Control	115.9 (27.2)	110.7 (36.5)	−5.2 (27.6)	0.153	0.498
	Intervention	129.0 (43.2)	120.4 (38.4)	−8.6 (22.7)	**0.019**	
	P1	0.073	0.203			
SGOT (iu/L)	Control	18.8 (7.8)	17.7 (6.6)	−1.2 (7.9)	0.212	0.461
	Intervention	18.8 (6.7)	15.9 (5.7)	−2.9 (5.1)	**0.003**	
	P1	0.999	0.162			
SGPT (iu/L)	Control	20.8 (11.1)	21.2 (12.1)	0.5 (8.6)	0.751	0.230
	Intervention	22.5 (12)	20.5 (10.1)	−2.1 (12)	0.169	
	P1	0.451	0.730			
γ-GT (iu/L)	Control	25.9 (16.4)	24.8 (19.2)	−1.1 (10.3)	0.564	0.373
	Intervention	27.8 (24.2)	24.4 (19.4)	−3.4 (15.1)	0.070	
	P1	0.639	0.923			
Urea (mg/dL)	Control	31.5 (8)	32.1 (9)	0.6 (8)	0.656	0.875
	Intervention	30.2 (6.9)	31.1 (10.4)	0.9 (10.5)	0.509	
	P1	0.367	0.587			
Uric acid- (mg/dL)	Control	5.14 (1.17)	5.39 (1.1)	0.25 (0.98)	0.086	0.103
	Intervention	5.29 (1.34)	5.2 (1.46)	−0.09 (1.07)	0.552	
	P1	0.545	0.476			
Creatinine (mg/dL)	Control	0.78 (0.2)	0.87 (0.49)	0.09 (0.45)	0.144	0.973
	Intervention	0.74 (0.13)	0.83 (0.41)	0.09 (0.42)	0.161	
	P1	0.247	0.642			

P1: *p*-value for group comparison, P2: *p*-value for time comparison, P3: Repeated measures ANOVA. Level of significance was set at 0.05. Values in bold point to significant differences. Effects reported include differences between the groups in the degree of change over the follow-up period; ALP: alkaline phosphatase, Fe: iron, CRP: C-reactive protein, LDH: lactate dehydrogenase. TC: total cholesterol, TG: triglycerides, HDL: high-density lipoprotein, LDL: low-density lipoprotein, SGOT: serum glutamic oxaloacetic transaminase, SGPT: serum glutamic pyruvic transaminase, γ-GT: γ-glutamyl transferase.

**Table 4 antioxidants-11-02113-t004:** Changes in 25(OH)D2 and 25(OH)D3 after 3 months in control vs. intervention group, by group.

		Pre	Post	Change		
	Group	Mean (SD)	Mean (SD)	Mean (SD)	P2	P3
25(OH)D3 ng/mL	Control	24.5 (10.4)	22.5 (10.2)	−2.0 (15.1)	0.293	0.358
	Intervention	24.2 (10.9)	24.6 (11.0)	0.4 (15.4)	0.804	
	P1	0.867	0.316			
25(OH)D2 ng/mL	Control	3.86 (4.85)	4.84 (5.07)	0.98 (6.3)	0.154	**0.014**
	Intervention	3.11 (3.78)	6.50 (4.69)	3.39 (4.2)	**<0.001**	
	P1	0.392	0.091			

P1: *p*-value for group comparison, P2: *p*-value for time comparison, P3: Repeated measures ANOVA. Level of significance was set at 0.05. Values in bold point to significant differences. Effects reported include differences between the groups in the degree of change over the follow-up period.

**Table 5 antioxidants-11-02113-t005:** Changes in anthropometric measurements after 3 months in control vs. intervention group.

		Pre	Post	Change		
	Group	Mean (SD)	Mean (SD)	Mean (SD)	P2	P3
Weight (kg)	Control	96.1 (16.7)	96.6 (16.8)	0.5 (10.9)	0.787	**0.031**
	Intervention	95.9 (20.3)	90.5 (21.3)	−5.4 (16)	**0.006**	
	P1	0.960	0.112			
Fat (kg)	Control	37.7 (12.6)	37.2 (11.7)	−0.6 (5.1)	0.427	0.186
	Intervention	39.6 (12.2)	37.7 (11.7)	−2 (5.3)	**0.009**	
	P1	0.447	0.826			
Fat (%)	Control	38.9 (8.2)	38.5 (8)	−0.4 (4.8)	0.590	**0.040**
	Intervention	41.4 (7)	39.0 (7.2)	−2.4 (4.8)	**0.001**	
	P1	0.108	0.757			
FFM (kg)	Control	58.4 (10.2)	57.8 (9.7)	−0.6 (2.6)	0.136	0.460
	Intervention	54.4 (12.3)	54.2 (12)	−0.2 (2.8)	0.651	
	P1	0.080	0.103			
TBW (kg)	Control	43.4 (7.2)	44.6 (7.7)	1.3 (5.3)	0.281	0.435
	Intervention	41.1 (10.4)	41.2 (9.2)	0.1 (10.7)	0.982	
	P1	0.219	**0.040**			
Visceral	Control	14.7 (5.2)	15 (5.7)	0.4 (2)	0.236	0.212
	Intervention	13.8 (4.7)	13.7 (5.3)	−0.2 (2.2)	0.560	
	P1	0.402	0.216			
ΒΜΙ (kg/m^2^)	Control	34.1 (6.2)	34.4 (6.2)	0.2 (4.1)	0.760	**0.026**
	Intervention	34.8 (6.7)	32.8 (6.8)	−2 (5.5)	**0.005**	
	P1	0.635	0.970			
WC (cm)	Control	112.4 (11.8)	112.2 (10.7)	−0.2 (4.6)	0.534	0.343
	Intervention	111.5 (15.1)	110.1 (14.5)	−1.4 (4.8)	**0.050**	
	P1	0.718	0.456			
HC (cm)	Control	117 (13)	116.5 (13)	−0.5 (5.1)	0.760	**0.041**
	Intervention	122.5 (20.1)	117.2 (14.3)	−5.2 (15.4)	**0.002**	
	P1	0.110	0.791			
SBP (mm/Hg)	Control	132.2 (16)	132.5 (14.7)	0.3 (14.8)	0.926	0.896
	Intervention	138.1 (19.5)	137.8 (21.5)	−0.3 (25.5)	0.926	
	P1	0.102	0.151			
DBP (mm/Hg)	Control	76.7 (9.8)	76.9 (9.2)	0.2 (6.2)	0.848	0.550
	Intervention	80.1 (11.4)	79.4 (9.2)	−0.7 (8.1)	0.513	
	P1	0.119	0.179			

P1: *p*-value for group comparison, P2: *p*-value for time comparison, P3: Repeated measures ANOVA. Level of significance was set at 0.05. Values in bold point to significant differences. Effects reported include differences between the groups in the degree of change over the follow-up period. FFM: free fat mass, TBW: total body water, BMI: body mass index, WC: waist circumference, HC: hip circumference, SBP: systolic blood pressure, DBP: diastolic blood pressure.

**Table 6 antioxidants-11-02113-t006:** Changes in inflammatory and oxidative stress markers after 3 months in control vs. intervention group.

		Pre	Post	Change		
	Group	Mean (SD)	Mean (SD)	Mean (SD)	P2	P3
Leptin (ng/mL)	Control	43.6 (45)	41.7 (42.8)	−1.9 (28)	0.604	0.501
	Intervention	54.8 (91.5)	56.6 (95.5)	1.8 (24.4)	0.664	
	P1	0.439	0.325			
IL-6 (pg/mL)	Control	2.8 (1.9)	2.9 (1.6)	0.1 (1.7)	0.702	**0.047**
	Intervention	3.2 (2.3)	2.7 (2)	−0.6 (1.6)	**0.016**	
	P1	0.353	0.452			
Adiponectin (μg/mL)	Control	12.4 (10.7)	10.9 (7.6)	−1.6 (8.4)	0.111	0.530
	Intervention	9.2 (9.5)	8.5 (7.9)	−0.7 (4.8)	0.475	
	P1	0.110	0.123			
MPO (ng/mL)	Control	125.6 (184.1)	162.6 (140.9)	37.0 (196)	0.137	0.900
	Intervention	178.6 (164.4)	211.1 (201.1)	32.5 (149.1)	0.190	
	P1	0.133	0.165			
oxLDL (U/L)	Control	81.29 (37.72)	80.88 (47.28)	−0.41 (40.05)	0.946	**0.020**
	Intervention	92.84 (53.12)	71.92 (44.09)	−20.92 (46.11)	**0.001**	
	P1	0.213	0.330			
TNF- α (pg/mL)	Control	1.21 (0.75)	1.19 (0.65)	−0.01 (0.96)	0.903	0.890
	Intervention	1.43 (0.64)	1.39 (0.59)	−0.04 (0.68)	0.750	
	P1	0.116	0.117			
8-isoprostane pg/mL)	Control	1893.1 (3753.8)	1912.7 (2878.1)	19.6 (3123.1)	0.872	0.856
	Intervention	1112.6 (1679.1)	1799.5 (2828.1)	686.9 (2957.6)	0.681	
	P1	0.779	0.518			

P1: *p*-value for group comparison, P2: *p*-value for time comparison, P3: Repeated measures ANOVA. Level of significance was set at 0.05. Values in bold point to significant differences. Effects reported include differences between the groups in the degree of change over the follow-up period. MPO: myeloperoxidase, oxLDL: oxidized LDL, TNF-α: tumor necrosis factor- α.

**Table 7 antioxidants-11-02113-t007:** Changes in QoL, depression, insomnia, self-esteem and physical activity.

		Pre	Post	Change		
	Group	Mean (SD)	Mean (SD)	Mean (SD)	P2	P3
AII	Control	6 (3.8)	5.5 (3.9)	−0.5 (2.8)	0.273	0.073
	Intervention	5.9 (4.1)	6.5 (4)	0.7 (3.7)	0.147	
	P1	0.831	0.201			
CESD-R-10	Control	18.9 (10.3)	17 (9.9)	−1.9 (8.0)	0.168	0.129
	Intervention	15.8 (9.7)	16.8 (11.3)	1 (10.6)	0.438	
	P1	0.123	0.909			
**Rosenberg Self-Esteem scale**	Control	31.1 (4)	31.4 (5.5)	0.2 (4.5)	0.729	0.840
	Intervention	30.8 (4.4)	30.8 (5)	0 (4.4)	0.951	
	P1	0.652	0.594			
PCS-12	Control	44.9 (9.2)	42.9 (9.2)	−2 (9.2)	0.139	**0.002**
	Intervention	43.9 (9.6)	48.1 (8.7)	4.1 (9.8)	**0.003**	
	P1	0.624	**0.004**			
MCS-12	Control	45.4 (10.2)	47.4 (9.9)	2.0 (7.0)	0.066	0.579
	Intervention	48.5 (8.6)	49.6 (9.6)	1.2 (8.1)	0.285	
	P1	0.113	0.267			
IPAQ-SF	Control	902.8 (916.3)	1326.9 (1685)	424.1 (1778.5)	0.240	0.293
(MET-min/week)	Intervention	2159.3 (2306.7)	2047.1 (2313.7)	−112.2 (3111.5)	0.755	
	P1	**0.001**	0.078			

P1: *p*-value for group comparison, P2: *p*-value for time comparison, P3: Repeated measures ANOVA. Level of significance was set at 0.05. Values in bold point to significant differences. Effects reported include differences between the groups in the degree of change over the follow-up period. AII: Athens Insomnia Scale, CESD-R: Center for Epidemiologic Studies Depression Scale Revised, PCS-12: Physical Composite Score, MCS-12: Mental Composite Score, IPAQ-SF: International Physical Activity Questionnaire (short form).

## Data Availability

All data are included within the article.

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
