# Peer review of "A Randomized Controlled Trial on Pleurotus eryngii Mushrooms with Antioxidant Compounds and Vitamin D2 in Managing Metabolic Disorders"

_antioxidants, 2022, doi:10.3390/antiox11112113_

Round 1

Author Response

uploaded in pdf

Reviewer 2 Report

The authors report a RCT on the effect of mushrooms  Pleurotus eryngii  (prepared snacks)  on metabolic parameters and vitamin D levels in metabolically unhealthy (obese)  patients. They find an increase in Vitamin D2 and an improvement of lipid profile.

The findings are of clinical significance, but the study period is rather short (3 months).  

Did the participants experience any side effects of the mushroom snack?

What are the costs of such a mushroom snack?

Author Response

uploaded

Reviewer 3 Report

Dear Editors

Dear Authors

 The presented manuscript entitled "A randomized controlled trial on Pleurotus eryngii mushrooms with antioxidant compounds and vitamin D2 in managing metabolic disorders" concerns the effects of the consumption of thermally processed P. eryngii on improving biochemical, inflammatory, oxidative stress and lifestyle parameters.

The research presented is well-planned, reliably described, and accompanied by numerous graphics.

As a reviewer, I have the following questions:

Due to the chemical composition of Pleurotus eryngii being well described in the scientific literature, the Authors should explain which specific compounds are responsible for the listed effects.

The chemical composition generally refers to untreated material. Do temperature-treated fruiting bodies have the same chemical composition?

I recommend this manuscript for publication in Antioxidants after these few explanations.

Author Response

uploaded
